# Research on the impact of digital economy on labor resource allocation: Evidence from China

Li Junfeng [1,2]*

1 School of Mathematics and Big Data, Chongqing University of Arts and Sciences, Chongqing, China,
2 Chongqing Key Laboratory of Statistical Optimization and Complex Data, Chongqing, China

* lijunfengswufe@163.com

**Data Availability Statement:** All relevant data are within the paper and its Supporting information files.

**Funding:** This article is funded by the Chongqing Social Science Planning Project "Research on the Impact of Digital Economy on Labor Factor

## Abstract

This paper establishes a coherent framework for delineating the nexus between the digital economy and the subjective efficacy of labor resource allocation. It elucidates the theoretical underpinnings of the digital economy's impact and its channel effects on the efficiency of labor allocation. Within the digital economy landscape, the phenomena of survivorship bias, digital divide, and algorithmic hegemony wield substantial sway over the efficiency of labor market allocation. Empirical analysis, conducted through a cross-sectional data model, validates the theoretical framework. The findings demonstrate that the digital economy markedly diminishes the subjective efficiency of labor allocation. Notably, this inhibitory effect is more pronounced among female workers, households with multiple residences, the non-unmarried demographic, and individuals over the age of 40, with the most pronounced effect observed among those aged over 60. In the examination of the causative mechanisms, it is discerned that the digital economy attenuates the subjective efficiency of labor allocation by workers through three conduits: alterations in social and economic status, shifts in living standards, and modifications in workplace comfort.

## Introduction

The digital economy in China has entered a new phase of rapid expansion and has emerged as a pivotal catalyst in the transformation of both traditional and emerging growth drivers, playing a vital role in fostering high-quality economic development. According to the "China Digital Economy Development White Paper (2021)" published by the China Academy of Information and Communications Technology, in 2020, the scale of China's digital economy soared to 39.2 trillion yuan, constituting a substantial 38.6% share of the GDP. Notably, the penetration rate of the digital economy in the service sector stood at an impressive 40.7%, markedly surpassing the figures of 21% in the industrial sector and 8.9% in the agricultural sector. Consequently, the digital economy holds profound implications for various factor markets, with a particularly significant impact on the labor market. In January 2022, the General Office of the State Council of China issued the "Comprehensive Reform Pilot Plan for Factor

Allocation and Response Strategies in Chongqing" (2022NDYB48), the Chongqing Education Commission Science and Technology Project "Measurement and Path of the Impact of Digital Economy on Innovation and Development in Chongqing" (KJQN202201317), and the Tower Plan of Chongqing University of Arts and Sciences (Introduction of Talents Project R2022SX05). Part of the data used in this paper is from the "2013 China General Social Survey" (CGSS) major project of the Chinese Academy of Social Sciences. The survey was conducted by the Institute of Sociology of the Chinese Academy of Social Sciences, and the project leader was Peilin Li. The author would like to thank the above institutions and their staff for their assistance in providing the data, and the author is solely responsible for the content of this paper.. The funders had role in the study design.

**Competing interests:** The authors have declared that no competing interests exist.

Marketization," wherein it explicitly underscored the imperative to "leverage the government's role, with a focus on dismantling institutional and systemic barriers that impede the organic and orderly flow of factors, thereby comprehensively enhancing the efficiency of factor coordination and allocation." Subsequently, in April 2022, the "Opinions of the Central Committee of the Communist Party of China and the State Council on Accelerating the Construction of a Unified National Market" (referred to as "Opinions" hereafter) were released, outlining the swift advancement of a unified national market as a primary goal, including the unification of factor markets. Furthermore, the report of the 19th National Congress of the Communist Party of China distinctly emphasized the marketization of factor allocation as one of the two pivotal focal points in the new phase of economic system reform in China.

Historical experience has shown that both the "Latin American trap" and the "middle-income trap" are closely related to the productivity loss caused by resource allocation [1], and the improvement of resource allocation efficiency mainly comes from the reduction of labor mobility barriers [2]. As an important factor market, the degree of marketization in the labor market has long been lower than that of the product market. According to the data published by the National Development and Reform Commission in 2017, the degree of marketization of product prices in China reached 97.01% in 2016, and the mechanism of market-determined prices has basically formed in the commodity market. However, the marketization process of factor markets still has a long way to go. The lag in the integration of factor markets is due to the distortion of local governments in the fiscal decentralization system towards factor markets. In the competition of regional economic development, local governments create realistic segmentation of the labor market through policies such as invisible household registration barriers, in order to achieve their short-term economic interests, which hinders the integration process of the labor market. Therefore, promoting the rational and orderly flow of labor, deepening household registration reform, is a new exploration to improve the factor market system and build a high-standard factor market system. The unification of the labor market can not only optimize the efficiency of labor resource allocation to release the demographic dividend, but also achieve the rational allocation of labor resources with other factor resources, thereby improving the total factor productivity of the economic system [3].

The digital economy provides the essential technological prerequisites for optimizing the allocation of labor resources. Since the onset of the COVID-19 pandemic in 2020, conventional cross-regional recruitment activities have experienced a near standstill. Nevertheless, employment services relying on digital information technology, such as online video conferences and job search platforms, have remarkably fulfilled the employment needs of both college students and enterprises. It is worth noting that the digital economy's contribution to the optimal allocation of labor resources extends beyond the establishment of efficient information matching platforms. As the digital economy has progressed, it has not only spurred the growth of the Chinese economy and refined resource allocation but has also dismantled regional market barriers, fostering the market-oriented evolution of labor factors across a broader spectrum.

In January 2022, the State Council issued the "Notice on Printing and Disseminating the 14th Five-Year Plan for the Development of the Digital Economy" (referred to as the "Notice" hereafter). This document outlined the guiding ideology, fundamental principles, and task objectives for the development of the digital economy during the 14th Five-Year Plan period. Over the past five years, China has achieved notable milestones in the industrialization and digitization of the digital industry. By 2020, the core industry's added value in China's digital economy accounted for 7.8% of GDP, with the digital economy's overall added value representing a substantial 38.6% of GDP, as reported in the "China Digital Economy Development White Paper" in 2021. The pervasive influence of the digital economy across various sectors

has transcended temporal and spatial limitations, facilitating the rapid dissemination of information throughout supply chains, financial networks, and service provision. This has led to the organic integration of the industrial Internet, service sector, and digital infrastructure, significantly enhancing the economic growth potential.

However, few domestic and foreign scholars have evaluated the impact of the digital economy on the labor market from the subjective experience of workers. The rise of the digital economy has restructured China's labor relations and embedded itself as a production factor in social production relations, fundamentally shaking up the labor market relations formed since the Third Industrial Revolution. Workers have transformed from operators of production machines to servants of social production. Traditional labor relations compromising, balancing, and fighting methods have been replaced by hidden algorithms. Workers' social factors such as technology, emotions, rationality, and collectiveness [4] have been submerged in the vast digital ocean.

The motivation underlying this study lies in the imperative to address the responsible innovation and research within the existing labor factor market in response to the risk challenges posed by the digital economy. This is essential to mitigate the impact of digital elements on the labor force. The innovation of this paper lies in two aspects. First, this paper examines, for the first time, the impact of China's early (2013) digital economy(The reason why the 2013 China Family Panel Studies (CFPS) database was chosen as the research sample in this article is that according to the "White Paper on China's Digital Economy Development 2021" report, the size of China's digital economy was 2.6 trillion yuan in 2005 and increased to 10 trillion yuan in scale after 2011 (accounting for 20.3% of GDP). Therefore, choosing the 2013 sample as the research object meets the behavioral characteristics of workers in the early stage of digital economic development) process on the subjective labor allocation efficiency of workers, supplementing the impact of China's digital economic development process on labor allocation efficiency observed from the perspective of workers. Second, this paper proposes that the early digital economy will reduce workers' subjective evaluation of labor market allocation efficiency and discover the transmission channels that affect labor market allocation efficiency.

## I. Literature review

In existing research, there are few studies that directly explore the relationship between the digital economy and labor resource allocation. This paper will review the relevant literature in two parts.

### (I) forms of labor market under the digital economy

The digital economy moves the factory to society, and the labor relations it creates are an innovation of the logic of capitalist exploitation. The process of work for laborers has shifted from the factory to society, thus starting the true complex labor factory of society. The flexibility of labor has increased, such as continuous online training, freelance work, the digital economy, and modes such as telecommuting [5]. However, the digital economy has not made traditional labor markets and labor processes disappear. Labor under the digital economy is neither free nor immaterial and is still given value by capital [6]. Most cognitive work in the digital era (50%-75% of non-programmable work) will continue to be reserved for human labor, while machines will handle routine programmable tasks. In the process of cooperation between humans and intelligent machines, human activities will greatly improve labor productivity when they dominate, while otherwise, it will reduce labor productivity levels [7].

## (II) Impact of digital economy on labor resource allocation

Overall, the development of the digital economy has improved the efficiency of labor resource allocation in China, with significant regional differences. The impact of digital economy development on labor resource allocation efficiency is most significant in the eastern region compared to the central and western regions. The intermediating mechanism variables that mitigate the mismatch of labor resources include flexible employment, employment platforms, and artificial intelligence [8–10].

At the same time, the rapid development of the digital economy has resulted in an increasing gap between labor supply and demand in the digital economy industry, with an evident trend of industry and regional concentration, which expands the digital divide between industries and regions [11]. The demand for labor in various industries will present a "polarization at both ends" pattern, reducing the demand for labor with high school and junior high school education and increasing the demand for labor with elementary school or below and college and above education. This is because sectors with a high level of digital economy require highly skilled labor, and as productivity and household income levels increase, the society's demand for service-oriented products has expanded, giving rise to a large number of positions that are not easily replaced by electronic devices [12]. This situation of medium-skilled labor being easily replaced by technology typically occurs in developed countries, where the cost of substituting medium-skilled workers is low, and wages are high. In developing countries, however, the cost of substituting medium-skilled workers is low, but wages are also low, resulting in medium-skilled workers being less likely to be replaced, while low-skilled labor is more easily replaced by technology [13].

In the short term, the digital economy will lead to a temporary increase in unemployment due to the mismatch between labor demand and supply structures, and in the long term, labor demand will gradually decrease due to technological substitution. Although technological progress eliminates "old professions" and creates new job fields, changing social relationships, its side effect is the reduction in labor demand, leading to an increase in unemployment rate [14, 15]. Nonetheless, the ramifications of digitalization on the labor market exhibit considerable disparity across distinct nations. Take, for instance, developing countries in the midst of transitioning from a "post-industrial economy" to a digital economy, where the impact of digital economic development on labor productivity often registers at zero or, in some instances, displays a negative trend. In stark contrast, developed countries experience a markedly positive effect from the digital economy, leading to substantial enhancements in personal income, elevated employment rates, and a reduction in labor market volatility [16, 17]. The development of the digital economy has caused industrialized countries to face problems such as educational mismatch, credential inflation, and job polarization [18].

Drawing from the synthesis of the aforementioned literature, it becomes evident that the digital economy engenders novel modes of employment, generally augmenting the efficiency of labor resource allocation. Nevertheless, for economies characterized by disparate economic underpinnings and developmental stages, the advent of negative effects cannot be discounted, particularly affecting low-skilled laborers and countries situated at a "post-industrialization level." Therefore, building upon the foundation laid by previous scholarly investigations, this paper embarks on a study focused on gauging the subjective assessments of Chinese workers concerning the nexus between the digital economy and labor resource allocation. This research endeavor aims to contribute to the existing body of knowledge in this field.

## II. Theoretical analysis

Chen Xiaohong et al. (2022) [19] conducted a comprehensive review and synthesis of the extant literature, revealing that the digital economy showcases distinctive attributes characterized by robust data support, seamless integrated innovation, and a culture of open sharing. These novel characteristics not only reshape labor dynamics but also exert a profound influence on the subjective perceptions of laborers, ultimately reshaping the assessment framework governing the allocation of labor resources.

### (i) The impact of survivorship bias on labor resource allocation

In theory, the digital economy has freed itself from the linear model of knowledge accumulation, research, and application, making knowledge sharing of products and services more flexible, resulting in the phenomenon of decentralized organizations and reduced organizational communication costs [2]. As a result, it has reduced the difficulty for workers to acquire and convert knowledge, allowing lower-level subjects to acquire and absorb knowledge and higher-level subjects to spread and disseminate knowledge [20]. Massive network knowledge dissemination platforms have resolved professional barriers between knowledge learning, and through a modular knowledge system(There are 200,000 packages available from the official Python repository alone, making it convenient for users to call upon them), have reduced learning difficulties, allowing ordinary workers to use digital technology(Artificial intelligence trainers are able to identify and analyze user corpora by training artificial intelligence, and even non-IT professionals can perform this job)to complete job tasks through a visualized platform, further improving the efficiency of labor resource allocation.

Nonetheless, it is essential to acknowledge that the survivorship bias inherent in the digital economy can also impede the efficiency of labor resource allocation. Survivorship bias pertains to the disparity between the chosen entities, often referred to as "survivors," and the broader population under observation due to selection criteria. As this selection process typically transpires unconsciously and can be easily overlooked, survivorship bias assumes particular significance for researchers engaged in fundamental theoretical inquiries. If the influence of survivorship bias is not properly accounted for or excluded, it can introduce ambiguity and uncertainty into research conclusions [21]. In the digital economy era, the survivorship bias effect has greatly amplified its power through the network, thus blurring the true supply and demand relationship of labor factor resources and reducing the awareness of one's own real economic status. Keywords such as "per capita monthly income exceeding ten thousand yuan," "success studies," "flaunting wealth," and "million-yuan-salary live streamers" in the online world have greatly impacted the minds of information receivers, causing people to ignore the logical relationship between input and output and pay more attention to the information released by these "survivors," ignoring the silent majority in the real world, resulting in a vague positioning of labor factor resources and falling into the "survivorship trap," which distorts information on compensation, working environment, and other information in labor relations. As the distortion intensifies, labor factor resources flow into the "survivors" field, causing waste of social labor resources, such as the hot live streaming industry, the proliferation of false high-salary job advertisements, and the resurgence of the "uselessness of education" argument.

This paper proposes hypothesis 1:

The survivorship bias effect of the digital economy will reduce laborers' positioning of their own real abilities.

## (ii) Impact of the digital divide on labor resource allocation

The data-driven nature of the digital economy is characterized by the efficient presentation and dissemination of massive amounts of information, with data resources becoming a key driver of labor resource allocation.

However, it's important to recognize that the digital divide can also emerge as a concern. The advancement of digital technology has effectively mitigated issues related to information asymmetry within the labor market, thereby reducing search and transaction costs for both labor supply and demand. By harnessing the inherent attributes of worker behavior, optimal employer-employee matching can be achieved, with job search platforms offering a wealth of employment opportunities and information, thereby diminishing job search expenses and enhancing the efficiency of supply-demand matching. Workers themselves are integral participants in the digital economy, continually enriching and enhancing employment-related information through their interactive networks. In doing so, they address gaps and pitfalls in the job market, optimizing resource allocation. The advent of workplace forums provides avenues for newcomers and secondary job seekers, who are on the cusp of entering the labor market, to make informed decisions and bridge information gaps within the labor market. The abundance of supply and demand information within the digital economy has spurred the emergence of niche demands, furnishing a marketplace for labor supply and demand with specialized skills. This evolution has gradually transformed numerous niche economies into mainstream economic drivers. The surge in live streaming and short-form videos has provided a platform for many workers possessing specialized skills to showcase their talents. This phenomenon has stimulated innovation and creativity among labor participants, reduced trial and error costs for individual entrepreneurs, and fostered a culture of widespread innovation and entrepreneurship. As a result, it has facilitated flexible employment opportunities for numerous workers.

However, it is crucial to acknowledge that not all individuals can fully reap the benefits of data support, primarily due to variances in the accessibility and proficiency in using digital technologies, such as the Internet. This issue has become particularly evident during the COVID-19 pandemic. Digital tools like health codes and travel passes have posed significant challenges for individuals who possess smart devices but lack the requisite information technology skills. This underscores the substantial impediments faced by the data support aspect of the digital economy in fully capitalizing on its advantages in the allocation of societal labor resources, especially in light of the digital divide. While a considerable portion of the population has access to the Internet, the process of career reorientation or entrepreneurship entails substantial risks when it involves searching for or establishing new ventures. The digital divide exacerbates the difficulties faced by entrepreneurs in swiftly and effortlessly accessing information resources, ultimately resulting in challenges related to securing financing [22].

While the data support aspect of the digital economy theoretically facilitates the efficient dissemination, self-optimization, and autonomous generation of labor market information, its practical reliance on robust digital infrastructure and workers proficient in information technology accentuates the real-world challenges posed by the digital divide. This digital divide can result in significant hardships for ordinary workers when navigating the digital economy, ultimately impeding the enhancement of efficiency in labor resource allocation within the digital economy and diminishing the overall well-being of the labor force. Therefore, this paper proposes hypothesis 2: The digital economy will cause a decline in workers' living standards due to the existence of the digital divide.

### (iii) Impact of algorithmic hegemony on labor resource allocation

Algorithmic hegemony denotes the conduct exhibited by data platform corporations, wherein they leverage the intricacies of algorithms and intentionally withhold transparency regarding their operations to establish an effective dominant position over platforms that are intertwined with the lives and work of individuals. In doing so, they encroach upon the interests of platform users.. In the era of the digital economy, the rapid emergence of open and collaborative network platforms, underpinned by elements like the Internet, big data, and blockchain, has facilitated the seamless integration of various facets of production, consumption, distribution, and services on these platforms. This integration has given rise to the platform economy, sparking innovative transformations across diverse realms of production, operations, and consumption. As a consequence, certain platforms have acquired both the incentive and the capability to persistently advance their business interests through the practice of algorithmic dominance.

The open and sharing platform economy has given rise to "de-labor relations" and reshaped the nature of labor-capital relations. "De-labor relations" refers to the removal of the constraints of employment or servitude, such as the use of labor contracts, de-employerization of employment, and concealment of the actual labor relationship [23]. In the context of the network economy, laborers represented by ride-hailing drivers, delivery couriers, and online broadcasters form "loose" labor relationships with platforms, freeing themselves from explicit supervision by their employers during the labor process and realizing a certain degree of labor freedom by combining their own human capital with the network platform [23], thereby enhancing their status as labor subjects. However, the labor process of these workers is still subject to covert surveillance under the "algorithmic logic". Because algorithmic logic has the inherent logical property of being "technically correct" [4], The convergence of fragmented labor relations serves to diminish the collective bargaining power of workers, resulting in a scenario where highly efficient laborers can exploit the algorithmic framework to maximize their own utility, while ordinary workers find themselves subjected to covert monitoring within the platform economy. For instance, food delivery platforms employ a "bottom-up competition" approach to continually push the delivery time limits for couriers, all the while precisely regulating their conduct through algorithms. Workers themselves may become ensnared within the "information cocoon" meticulously designed by platform companies, restricting their exposure to limited information channels akin to cocoons. This, in turn, hinders their ability to engage in independent thinking, self-improvement, and the opportunity to accumulate human capital through on-the-job learning. Consequently, it ultimately leads to diminished job satisfaction.

Therefore, this article proposes hypothesis 3: the algorithmic hegemony of the digital economy will reduce the satisfaction of laborers in the labor process.

## III. Empirical study design, data and indicator metrics

### (i) Model setting

According to the results of the theoretical analysis, the digital economy will change the working environment of labor subjects and affect the level of labor factor allocation, therefore, the following cross-sectional data model is constructed in this paper to empirically investigate the findings of the theoretical analysis.

$$lfa_{ij} = \alpha_0 + \alpha_1 de_j + Z_{ij}{'}\alpha_3 + \varepsilon_i \tag{1}$$

Where $lfa_{ij}$ is the level of marketed labor allocation for the $i$ sample in $j$ province, $de_j$ is the level

of digital economy variable in $j$ province, the control variable associated with $Z_{ij}{}'$, and $\varepsilon_i$ is the nuisance term.

## (ii) Data variables

This paper utilizes data from the 2013 Chinese Social Survey (CSS), which covers 31 provinces, autonomous regions, and municipalities, including 151 districts/counties and 604 villages/neighborhood committees, and involves visiting 7,000 to 10,000 households per survey(The data used in this paper are partly from the 2013 Comprehensive Survey on Social Conditions in China, a major project of the Chinese Academy of Social Sciences. The survey was executed by the Institute of Sociology, Chinese Academy of Social Sciences, and the project leader was Li Peilin. The authors would like to thank the above-mentioned institutions and their personnel for providing data assistance, and the authors are responsible for the content of this thesis). This database provides comprehensive and scientific information on long-term longitudinal surveys of labor and employment of the general public nationwide, which serves as a solid foundation for social science research.

The dependent variable in this study pertains to the degree of labor factor marketization. To gauge this, survey data from the CSS database were utilized, specifically the responses to the question "How fair do you think the current job and employment opportunities are in society?" This variable is used as a proxy for the level of labor factor marketization, with responses categorized as follows: "1" denoting "very unfair," "2" indicating "unfair," "3" representing "fairly fair," and "4" signifying "very fair." Responses marked as "8," indicating "hard to say," were excluded from the sample data for this study. It is worth noting that numerous scholars have undertaken related research endeavors on the marketization of labor factor resources. In their research, Chen Yongwei and Hu Weimin (2011) [24] employed a method to calculate the degree of labor factor mismatch. This involved using the actual proportion of labor allocated to industry i as the numerator, and the theoretical proportion of labor allocated to industry i under efficient labor factor allocation as the denominator. The resulting ratio gauged the degree of labor factor mismatch in industry i. However, it's important to note that this method primarily assesses relative distortion rather than absolute distortion. The National Academy of Development and Strategy at Renmin University of China released an annual report on China's labor marketization index from 2010 to 2016. This index assesses two critical aspects of China's labor market: the market segmentation stemming from household registration and system factors (such as the openness of household registration and the proportion of employees in state-owned enterprises), as well as labor pricing (minimum wage and salaries of employees in state-owned enterprises). Meanwhile, Beijing National Economic Research Institute have been calculating China's labor force marketization degree since the publication of the "China Marketization Index Report" in 1997. The indicators they employ primarily focus on the supply of technical and managerial personnel, skilled workers, and the ratio of permanent residents to registered residents. However, these indicators have their limitations, often operating at the provincial or municipal level. Moreover, they tend to emphasize distortions arising from state-owned enterprises and household registration in the labor market while overlooking the subjective perceptions of residents concerning the level of labor factor resource allocation. In contrast, the present study leverages residents' subjective assessments of job and employment opportunities from the CSS survey data as a proxy for labor factor marketization. This approach avoids the issue of data aggregation and combines the subjective evaluations of workers from diverse backgrounds with the degree of labor factor allocation marketization, thus offering a more precise measurement of labor resource configuration from the perspective of "talent optimization."

The core independent variable is the inter-provincial digital economy index, which uses the "China Inter-provincial Digital Economy Development Level" data calculated by Wang et al. (2021) [25]. This database is based on the "connotation of digital economy, focusing on the conditions, applications and environment of digital economy, and comprehensively building a digital economy index system". It sets four indicators: digital economy development carriers, digital industrialization, industrial digitalization, and digital economic development environment, with 30 sub-indicator variables covering the measured data of the digital economy development level of 30 provinces from 2013 to 2018.

## (iii) Descriptive statistics

At the same time, this study uses household size, age of respondents, educational level, marital status, household registration status, current employment status (employed or unemployed), wage income, job skills requirements, job unit nature (party and government, state-owned enterprise, private enterprise), number of family houses, family income and expenditure surplus or deficit, internet use, and subjective socioeconomic status as control variables, to control for the influence of factors other than the core independent variable on the research conclusions as much as possible. The specific variable descriptions and descriptive statistics results can be found in Tables 1 and 2.

## (iv) Empirical study

**1. Regression using instrumental variables method.** The digital economy may be an endogenous choice for labor market allocation, and the measurement of digital economy level may also be subject to measurement error due to methodological differences, as well as omitted variable bias in the model. The benchmark model is difficult to simultaneously address the issues of bidirectional causality, measurement error, and omitted variables, resulting in inconsistent OLS estimates and an inability to converge to the true parameter level. Therefore, this paper attempts to mitigate the endogeneity problem in the model through instrumental variable methods. Following the approach of Huang Qunhui et al. (2019) and Guo Dongjie et al. (2022) [26, 27], the number of fixed telephone lines per capita in each region in 1988 (fp) is used as an instrumental variable for the digital economy, as the digital economy is an economic model developed on the basis of new communication technologies, which is a

**Table 1. Variable descriptions.**

| Variables | Variable Description |
|-----------|---------------------|
| la | Labor allocation efficiency |
| de | Digital Economy |
| age | Respondents' age |
| edu | Education level |
| ws | Working situation |
| wage | Wages and Salaries |
| js | Job Skill Requirements |
| nw | Nature of work unit |
| ie | Surplus or deficit of household income and expenditure |
| int | Internet Skills |
| M1 | Socio-economic status |
| M2 | Changes in living standards |
| M3 | Work comfort |

**Table 2. Descriptive statistics.**

| Variables | Obs | Mean | Std. Dev. | Min | Max |
|---|---|---|---|---|---|
| la | 2024 | 2.385 | .67 | 1 | 4 |
| de | 2024 | .147 | .093 | .013 | .35 |
| age | 2024 | 38.705 | 10.708 | 18 | 69 |
| edu | 2024 | 4.881 | 3.048 | 1 | 98 |
| ws | 2024 | 1.059 | .236 | 1 | 2 |
| wage | 2024 | 2929.545 | 4258.58 | 0 | 90000 |
| js | 2024 | 3.184 | 1.182 | 1 | 5 |
| nw | 2024 | 4.753 | 2.761 | 1 | 12 |
| ie | 2024 | 1.983 | 1.027 | 1 | 8 |
| int | 2024 | 1.375 | .484 | 1 | 2 |
| M1 | 2024 | 3.527 | .853 | 1 | 5 |
| M2 | 2024 | 2.171 | .896 | 1 | 5 |
| M3 | 2020 | 3.11 | .911 | 1 | 5 |

continuation of the application of traditional communication technologies such as fixed telephones. The development of Internet technology is based on fixed telephone technology, and regions with a high fixed telephone penetration rate in history are also likely to have a high Internet penetration rate. The layout of fixed telephones affects the early access to the Internet, which in turn affects the residents' technology use and habits [26].

Table 3 reports the results of the instrumental variable regression. Both the Kleibergen-Paap rk LM statistic and the Kleibergen-Paap rk Wald F statistic reject the null hypothesis for instrument invalidity, while the Hansen J statistic for overidentification only has one instrument and meets the exact identification condition, indicating that the instrumental variable used in this study is valid and there is no weak instrument problem, and the null hypothesis of all instruments being exogenous cannot be rejected.

Model (1) in Table 3 reports the parameter estimates of the core explanatory variable la. After the instrumental variable regression, the parameter of la is significantly negative at the 5% level, indicating that the digital economy significantly reduces the subjective perception of labor allocation efficiency for workers. In Model (2), the parameter of the instrument variable fp is significantly positive at the 1% level, satisfying the requirement of the correlation between the instrument variable and the endogenous variable.

**2. Heterogeneity study.** Based on the instrumental variable regression analysis in the previous section, the overall impact of the digital economy on labor allocation efficiency has been revealed. However, this impact may have heterogeneity. Therefore, this study investigates the heterogeneity of the impact of the digital economy on labor allocation efficiency from four dimensions: gender, housing ownership, marital status, and age.

Table 4, models (1) and (2), report on the impact of gender heterogeneity on the research findings. For the female group, the impact parameter of the digital economy on labor allocation efficiency is significantly negative, while for the male group, this parameter is also negative but not significantly different from zero. This indicates that the negative impact of the digital economy on the efficiency of female labor market allocation is deeper. This is because women, as a group with lower risk preferences, tend to engage in stable jobs [28], and the digital economy has had an impact on the existing labor market form, breaking the inherent work patterns, thereby reducing the subjective labor allocation efficiency of the female group.

**Table 3. Regression results using instrumental variables method.**

| | 2SLS | First stage regression results |
|---|---|---|
| | la | de |
| | (1) | (2) |
| de | -0.669** | |
| | (0.340) | |
| fp | | 9.484*** |
| | | (24.26) |
| age | -0.003 | -0.001** |
| | (0.002) | (-2.58) |
| edu | -0.014* | -0.001 |
| | (0.007) | (-0.82) |
| ws | -0.002 | -0.010 |
| | (0.061) | (-1.41) |
| wage | 0.000 | 0.000 |
| | (0.000) | (1.31) |
| is | -0.016 | 0.003* |
| | (0.016) | (1.68) |
| nw | -0.004 | 0.001* |
| | (0.006) | (1.77) |
| ie | -0.033** | 0.002 |
| | (0.015) | (0.91) |
| int | 0.042 | -0.017*** |
| | (0.042) | (-3.50) |
| Constant | 2.720*** | 0.133*** |
| | (0.143) | (9.42) |
| Phase I F-statistic | 588.477*** | |
| Kleibergen-Paap rk LM statistic | 348.017*** | |
| Kleibergen-Paap rk Wald F statistic | 588.467*** | |
| Hansen J statistic | Adequate identification | |
| Observations | 1,991 | 1,991 |
| Adjusted R-squared | -0.009 | |

Note: The values in parentheses represent clustered robust standard errors. This applies to all subsequent mentions as well.

Models (2) and (3) report on the impact of the number of homes owned on the research findings. The inhibitory effect of the digital economy on the labor allocation efficiency of households with multiple homes is higher than that of households with fewer or no homes, possibly because: firstly, owning homes can occupy family assets and even lead to family debts. Many families become "mortgage slaves" due to housing loans, and such families will avoid risk investment behavior to reduce overall family risk [29]. When the digital economy has new impacts on the labor market, families with multiple homes will be more sensitive to the impact; secondly, the number of properties can to some extent make up for class identity formed by factors such as income, occupation, and education [30]. As a new phenomenon, the digital economy has impacted this class identity at the labor relations level, making multi-home households more anxious when facing new challenges in the labor factor market.

**Table 4. Heterogeneity study.**

| | Women | Male | 2 or more housing units | 1 housing unit or less | Non-Unmarried | Unmarried | 20~40 years old | 40~60 years old | over60 years old |
|---|---|---|---|---|---|---|---|---|---|
| | (1) | (2) | (3) | (4) | (5) | (6) | (7) | (8) | (9) |
| de | -0.917* | -0.512 | -1.685** | -0.357 | -0.934** | 0.589 | -0.231 | -0.951* | -4.211** |
| | (0.515) | (0.452) | (0.703) | (0.389) | (0.380) | (0.741) | (0.484) | (0.519) | (1.876) |
| age | -0.006** | -0.001 | 0.002 | -0.004** | -0.002 | -0.008 | -0.005 | 0.004 | -0.027 |
| | (0.003) | (0.002) | (0.003) | (0.002) | (0.002) | (0.007) | (0.004) | (0.004) | (0.042) |
| edu | -0.040*** | -0.011* | -0.043** | -0.010* | -0.012* | -0.028 | -0.030*** | -0.032** | -0.004 |
| | (0.015) | (0.006) | (0.022) | (0.006) | (0.007) | (0.021) | (0.012) | (0.016) | (0.003) |
| ws | -0.076 | 0.071 | 0.067 | 0.009 | -0.066 | 0.242* | -0.014 | -0.016 | 0.459** |
| | (0.091) | (0.082) | (0.174) | (0.067) | (0.068) | (0.131) | (0.080) | (0.104) | (0.187) |
| wage | 0.000 | 0.000 | -0.000 | 0.000 | 0.000 | -0.000 | 0.000 | 0.000 | -0.000 |
| | (0.000) | (0.000) | (0.000) | (0.000) | (0.000) | (0.000) | (0.000) | (0.000) | (0.000) |
| is | -0.040 | -0.011 | -0.017 | -0.017 | -0.016 | -0.022 | -0.041* | -0.006 | -0.060 |
| | (0.026) | (0.019) | (0.039) | (0.017) | (0.016) | (0.046) | (0.024) | (0.023) | (0.065) |
| nw | -0.011 | -0.001 | 0.017 | -0.008 | -0.005 | 0.002 | -0.004 | -0.011 | -0.001 |
| | (0.011) | (0.007) | (0.016) | (0.007) | (0.007) | (0.019) | (0.010) | (0.009) | (0.028) |
| ie | 0.003 | -0.066*** | -0.054 | -0.028* | -0.046** | -0.009 | -0.016 | -0.053** | -0.186* |
| | (0.020) | (0.022) | (0.034) | (0.017) | (0.018) | (0.024) | (0.019) | (0.026) | (0.107) |
| int | -0.014 | 0.051 | 0.065 | 0.045 | 0.043 | 0.052 | 0.075 | -0.007 | -0.069 |
| | (0.066) | (0.055) | (0.097) | (0.046) | (0.043) | (0.184) | (0.067) | (0.060) | (0.194) |
| _cons | 3.216*** | 2.551*** | 2.711*** | 2.699*** | 2.827*** | 2.441*** | 2.832*** | 2.655*** | 5.059* |
| | (0.250) | (0.171) | (0.350) | (0.151) | (0.156) | (0.391) | (0.237) | (0.300) | (2.788) |
| N | 805 | 1186 | 383 | 1608 | 1686 | 305 | 1020 | 886 | 63 |
| adj. $R^2$ | -0.020 | -0.001 | -0.027 | -0.000 | -0.015 | -0.005 | -0.000 | -0.016 | -0.255 |

Note: The regression results of the first step of instrumental variables in the model and the correlation test of instrumental variables are omitted.

Model (4) and (5) report the impact of marital status on the research findings. Compared with unmarried individuals, the negative effect of digital economy on labor market allocation efficiency is more significant for non-unmarried groups. One possible reason is that non-unmarried individuals face greater life pressure and are therefore more sensitive to new changes in the labor market.

Models (5), (6), and (7) report the impact of age groups on the research findings. Among individuals over 40 years old, the negative effect of digital economy on labor market allocation efficiency is significant, particularly for those over 60 years old. However, this negative effect is not significant among individuals aged between 20 and 40. This is mainly due to the fact that younger people have stronger learning abilities and opportunities for trial and error to cope with the impact of digital economy on the labor market. Individuals over 40 years old, especially the elderly, have difficulty in effectively coping with the impact of digital economy on their labor market.

**3. Mechanism research.** According to the instrumental variables method, it is concluded that the digital economy significantly reduces the subjective perception of labor allocation efficiency, which is inconsistent with the conclusions of previous literature. In order to explore the mechanism by which the digital economy affects labor allocation efficiency, this paper analyzes the mechanism pathways based on theoretical analysis, and constructs a mediation effect model by analyzing three pathways: "socioeconomic status" (M1), "changes in living standards"

(M2), and "job comfort" (M3).

$$m_{ij} = b_0 + b_1 de_j + Z_{ij}'b_3 + \varepsilon_i \tag{2}$$

$$lfa_{ij} = c_0 + c_1 m_{ij} + c_2 de_j + Z_{ij}'c_3 + \varepsilon_i \tag{3}$$

Table 5 presents the regression results with M1 as the mediator in Models (1) and (2).

**Table 5. Mechanism analysis.**

|  | 2SLS | 2SLS | 2SLS | 2SLS | 2SLS | 2SLS |
|---|---|---|---|---|---|---|
|  | M1 | la | M2 | la | M3 | la |
|  | (1) | (2) | (3) | (4) | (5) | (6) |
|  | g1 | g2 | g3 | g4 | g5 | g6 |
| de | 1.253*** | -0.594* | 1.474*** | -0.532 | -0.977* | -0.563* |
|  | (0.431) | (0.341) | (0.517) | (0.335) | (0.505) | (0.338) |
| M1 |  | -0.059*** |  |  |  |  |
|  |  | (0.019) |  |  |  |  |
| M2 |  |  |  | -0.092*** |  |  |
|  |  |  |  | (0.018) |  |  |
| M3 |  |  |  |  |  | 0.093*** |
|  |  |  |  |  |  | (0.017) |
| age | -0.002 | -0.003 | 0.007*** | -0.002 | 0.003 | -0.003* |
|  | (0.002) | (0.002) | (0.002) | (0.002) | (0.002) | (0.002) |
| edu | -0.008 | -0.014* | 0.016*** | -0.012* | -0.007 | -0.013* |
|  | (0.016) | (0.008) | (0.006) | (0.007) | (0.006) | (0.007) |
| ws | -0.007 | -0.003 | -0.122 | -0.013 | -0.069 | 0.005 |
|  | (0.079) | (0.060) | (0.077) | (0.061) | (0.093) | (0.063) |
| wage | -0.000 | 0.000 | 0.000 | 0.000 | -0.000 | 0.000 |
|  | (0.000) | (0.000) | (0.000) | (0.000) | (0.000) | (0.000) |
| is | 0.112*** | -0.009 | 0.045** | -0.012 | -0.100*** | -0.005 |
|  | (0.020) | (0.016) | (0.020) | (0.016) | (0.021) | (0.016) |
| nw | 0.011 | -0.003 | -0.015** | -0.005 | 0.002 | -0.005 |
|  | (0.008) | (0.006) | (0.008) | (0.006) | (0.008) | (0.006) |
| ie | 0.096*** | -0.028* | 0.128*** | -0.022 | -0.020 | -0.031** |
|  | (0.020) | (0.015) | (0.022) | (0.015) | (0.020) | (0.015) |
| int | 0.117** | 0.049 | -0.087 | 0.034 | 0.030 | 0.041 |
|  | (0.055) | (0.042) | (0.054) | (0.042) | (0.058) | (0.042) |
| _cons | 2.737*** | 2.882*** | 1.531*** | 2.859*** | 3.558*** | 2.382*** |
|  | (0.224) | (0.157) | (0.171) | (0.144) | (0.180) | (0.154) |
| Phase I F-statistic | 588.580*** | 584.520*** | 588.580*** | 581.160*** | 580.530*** | 580.67*** |
| Kleibergen-Paap rk LM statistic | 348.083*** | 349.647*** | 348.083*** | 343.211*** | 346.618*** | 347.761*** |
| Kleibergen-Paap rk Wald F statistic | 588.583*** | 584.517*** | 5883583*** | 581.156*** | 580.534*** | 580.668*** |
| Hansen J statistic | Adequate identification | Adequate identification | Adequate identification | Adequate identification | Adequate identification | Adequate identification |
| N | 1994 | 1991 | 1994 | 1991 | 1990 | 1987 |
| adj. $R^2$ | 0.057 | -0.002 | 0.026 | 0.009 | 0.004 | 0.009 |

Note: The regression results of the first step of the instrumental variables in the model are omitted.

Model (1) shows that the increase in the level of the digital economy (de) significantly increases the value of the social and economic status variable (M1) (M1 is a contrarian indicator, with higher M1 values indicating that "the respondent perceives their own socioeconomic status to be relatively lower in the local area."), indicating that the improvement in the level of the digital economy significantly reduces respondents' self-positioning in terms of social and economic status. Model (2) indicates that the increase in the social and economic status variable (M1) significantly reduces the subjective perception of labor allocation efficiency, that is, the decrease in respondents' self-positioning in terms of social and economic status reduces the subjective perception of labor allocation efficiency. Models (1) and (2) jointly demonstrate that the improvement in the level of the digital economy reduces labor allocation efficiency (subjective perception) by reducing respondents' self-positioning in terms of social and economic status through the mediating channel, thus verifying Hypothesis 1.

Models (3) and (4) report the regression results with M2(M2 is a contrarian indicator, with higher M2 value indicating that " a decline in your standard of living compared to five years ago.") as the mediator. Model (3) shows that the increase in the level of the digital economy (de) significantly increases the value of the variable measuring changes in living standards (M2), indicating that the improvement in the level of the digital economy significantly reduces respondents' living standards compared to five years ago. Model (4) indicates that the increase in the variable measuring changes in living standards (M2) significantly reduces the subjective perception of labor allocation efficiency, that is, the decrease in respondents' living standards reduces the subjective perception of labor allocation efficiency. Models (3) and (4) jointly demonstrate that the improvement in the level of the digital economy reduces labor allocation efficiency through the mediating channel of respondents' living standards, thus verifying Hypothesis 2.

Models (5) and (6) report the regression results with M3(M3 is a positive indicator, and a higher value of M3 indicates "a higher frequency of experiencing positive emotions and enjoyment during work.") as the mediator. Model (5) shows that the increase in the level of the digital economy (de) significantly reduces the value of the variable measuring work comfort during working hours (M3), indicating that the improvement in the level of the digital economy reduces work comfort. Model (6) indicates that the increase in the variable measuring work comfort during working hours (M3) enhances the subjective perception of labor allocation efficiency. Models (5) and (6) jointly demonstrate that the improvement in the level of the digital economy reduces labor allocation efficiency through the mediating channel of work comfort, thus verifying Hypothesis 3.

## IV. Conclusions of the study

This paper has established a coherent conceptual framework elucidating the relationship between the digital economy and subjective labor allocation efficiency. At the theoretical level, it has expounded upon the impact and channels through which the digital economy influences labor allocation efficiency. Furthermore, this theoretical framework has been subjected to empirical scrutiny utilizing cross-sectional data models.The study's findings underscore the substantial reduction in subjective labor allocation efficiency prompted by the digital economy. Notably, this inhibitory effect is more pronounced among female workers, households with multiple residences, individuals who are not married, and those aged over 40, with the most significant impact observed among individuals over 60 years old. In the investigation of the causal mechanisms, the research discerns that the digital economy diminishes workers' subjective labor allocation efficiency via three avenues: changes in their social and economic status, shifts in living standards, and alterations in workplace comfort.

This paper undertakes an initial exploration into the early repercussions of the digital economy on labor allocation efficiency, with the intent of shedding light on the adverse effects experienced by specific segments of the population. Several key insights emerge from this examination:

Firstly, the "information cocoon" engendered by the digital economy can result in a lowering of individuals' perceptions regarding their own real circumstances, potentially ensnaring them in cognitive traps. Consequently, it becomes imperative for all sectors of society to elucidate the actual state of the digital economy and enhance workers' comprehension of labor relations in the digital era comprehensively. Secondly, the "digital divide" precipitated by the digital economy can lead to the exclusion of certain individuals from the digital dividend due to their reluctance or inability to engage with digital tools. In response, society as a whole must intensify efforts to bolster digital infrastructure, ensuring that workers have access to high-quality network information services. Addressing the "last mile" of digital economic development is paramount.

Thirdly, within the context of algorithmic hegemony in the digital economy era, the clandestine influence of algorithmic power continues to erode the legitimate rights and interests of workers. Workers themselves may find it challenging to mount a unified front against algorithmic hegemony as they become entrapped within the sphere of algorithmic control. Society should undertake a reevaluation of the societal value of algorithms, seeking to empower workers with greater autonomy and initiative in their engagement with the confluence of logical reasoning and algorithmic logic [4]. Lastly, it is essential to consider the subjective perceptions of specific societal groups, and when deemed necessary, implement targeted social relief measures to address their unique needs and challenges.

Certainly, this study has concentrated on the efficiency of labor resource allocation during the nascent phase of China's digital economy. As China's digital economy continues to evolve and mature, it will be crucial to remain vigilant on this matter and assess whether the conclusions drawn in this study undergo transformations in response to the ongoing development of the digital economy.

## Supporting information

**S1 File.**
(DTA)

**S2 File.**
(DO)

## Author Contributions

**Conceptualization:** Li Junfeng.

**Data curation:** Li Junfeng.

**Formal analysis:** Li Junfeng.

**Funding acquisition:** Li Junfeng.

**Investigation:** Li Junfeng.

**Methodology:** Li Junfeng.

**Project administration:** Li Junfeng.

**Resources:** Li Junfeng.

**Software:** Li Junfeng.

**Supervision:** Li Junfeng.

**Validation:** Li Junfeng.

**Visualization:** Li Junfeng.

**Writing – original draft:** Li Junfeng.

**Writing – review & editing:** Li Junfeng.

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
