## [Decision Letter · Decision Letter 0]

8 May 2023

PONE-D-23-09598Research on the Impact of Digital Economy on Labor Resource Allocation: Evidence from ChinaPLOS ONE

Dear Dr. junfeng,

Thank you for submitting your manuscript to PLOS ONE. After careful consideration, we feel that it has merit but does not fully meet PLOS ONE’s publication criteria as it currently stands. Therefore, we invite you to submit a revised version of the manuscript that addresses the points raised during the review process.

We look forward to receiving your revised manuscript.

Kind regards,

Shi Yin

Academic Editor

PLOS ONE

Journal Requirements:

Reviewers' comments:

Reviewer's Responses to Questions

**Comments to the Author**

1. Is the manuscript technically sound, and do the data support the conclusions?

Reviewer #1: Yes

Reviewer #2: Yes

2. Has the statistical analysis been performed appropriately and rigorously? 

Reviewer #1: Yes

Reviewer #2: Yes

3. Have the authors made all data underlying the findings in their manuscript fully available?

Reviewer #1: No

Reviewer #2: Yes

4. Is the manuscript presented in an intelligible fashion and written in standard English?

Reviewer #1: No

Reviewer #2: Yes

5. Review Comments to the Author

Reviewer #1: This paper focuses on the Impact of Digital Economy on Labor Resource Allocation. The topic is of practical significance. However, the level of this version needs to be improved, and the research logic and theoretical logic are not clear enough. It is suggested that the manuscript should be major revised and reviewed again. Some suggestions and comments are as follows：

Reviewer #2: This paper constructs a logical framework between the digital economy and subjective labor allocation efficiency. It theoretically explains the impact and channels of the digital economy on labor allocation efficiency, and finds that the digital economy significantly reduces workers' subjective perception of labor allocation efficiency. The paper observes the impact of China's digital economy development process on labor allocation efficiency from the perspective of workers, which has important practical significance.The paper has certain innovation and practical meaning. However, there are still several issues that need to be addressed:

1.The titles of every section need to be improved to be formal.

2.The variables "socioeconomic status" (M1), "changes in living standards" (M2), and "job comfort" (M3) should be given specific explanations and whether they are positive or negative indicators.

3.The footnote "D1 is a reverse indicator, with higher D1" is incorrect and should be corrected to M1.

4.It would be helpful to include some more recent research findings, which would enhance the forward-looking nature of the article.

5.In the conclusion section, some concrete recommendations could be given to address these challenges.

6.The figures and tables in the paper could be redesigned to make them more clear and easy to understand.

6. PLOS authors have the option to publish the peer review history of their article (what does this mean?). If published, this will include your full peer review and any attached files.

Reviewer #1: No

Reviewer #2: No

---

## [Author Response · Author response to Decision Letter 0]

21 Jul 2023

Response to reviewer #1:

1.According to the literature by Wen Yanbing et al. (2022), both the 'Latin American trap' and the 'middle-income trap' are closely related to productivity losses caused by resource allocation. Additionally, the development of the digital economy provides the technological conditions for optimizing the allocation of labor resources. These factors serve as reasons to investigate the relationship between the digital economy and the allocation of labor resources. Furthermore, in order to enhance the persuasiveness of the introduction section, this paper includes information on the development of the digital economy in China. For instance,

“Digital economy has entered a new stage of rapid development in China and has become an important engine for the transformation of old and new drivers of growth as well as for promoting high-quality economic development. According to the "China Digital Economy Development White Paper (2021)" released by the China Academy of Information and Communications Technology, the scale of China's digital economy reached 39.2 trillion yuan in 2020, accounting for a high proportion of 38.6% of the GDP. The digital economy penetration rate in the service sector was 40.7%, far higher than the 21% in the industrial sector and 8.9% in the agricultural sector. Therefore, the digital economy has profound implications for factor markets, especially the labor market.”

Regarding the limitations of existing research, this paper addresses them in the literature review section by stating, ' based on previous research by scholars, this paper studies the subjective evaluation of Chinese workers on the relationship between the digital economy and labor resource allocation, in order to improve existing research.' Existing studies have focused on the impact of the digital economy on labor demand and structure, while lacking a subjective evaluation of the relationship between the digital economy and labor resource allocation.

China has carried out extensive practices in the development of the digital economy. As a developing country, its practices in digital economy development hold significant practical significance. Therefore, this paper starts from the reality in China and considers the subjective evaluation of the relationship between Chinese workers and the digital economy in terms of labor resource allocation.

2.The literature review section has already covered the forms of the labor market under the digital economy and the research on the impact of the digital economy on labor allocation. As for the "positive and negative impacts" mentioned by the reviewers, they have also been addressed in the literature review. However, this paper does not intend to structure the literature review as a combination of "positive and negative impacts" because the development of the digital economy is a prevailing trend, and discussing these aspects does not hold much significance.

3. The conclusion that ' These new features not only change labor relations but also profoundly affect the subjective perception of laborers, and reconstruct the evaluation system of labor resource allocation.' is derived from the research conducted by Chen Xiaohong et al. (2022). The data-driven nature of the digital economy is characterized by the efficient presentation and dissemination of massive amounts of information, with data resources becoming a key driver of labor resource allocation. However, it can also create the problem of the digital divide…… cultivates a system of mass innovation and entrepreneurship, achieving flexible employment for many workers." These aspects originate from the background reality in China, and they are observable phenomena, whether it be professional job search websites or popular platforms such as Douyin and Kuaishou for short videos. Therefore, there is no need to specifically seek literature support for these statements.

4. At present, there are no available papers that specifically focus on researching laborers' subjective evaluations in relation to the digital economy. Therefore, it is currently difficult to make direct comparisons with peer-reviewed studies in the empirical research section.

5. Note: (The values in parentheses represent clustered robust standard errors. This applies to all subsequent mentions as well.) have been added in the table.

6. For the authors, it is challenging to propose specific solutions as the transformation of the labor market brought about by the digital economy inevitably has some negative impacts on certain groups. Currently, it is difficult to address these issues solely through policy measures.

7. There is a limited amount of literature from foreign countries that focuses on labor resource allocation issues under the digital economy in China. As for the abundance of Chinese literature, it is mainly due to domestic scholars being the main contributors in researching labor resource allocation in China. It is challenging to find more valuable information to support the arguments of this paper in foreign literature.

8. This point has been revised accordingly, and appropriate annotations have been made in the text.

Response to reviewer #2:

1. The title has been reviewed.

2. M1 is a contrarian indicatorreverse indicator, with higher DM1 values indicating that "the respondent perceives their own socioeconomic status to be relatively lower in the local area."

M2 is a contrarian indicator, with higher M2 value indicating that " a decline in your standard of living compared to five years ago."

M3 is a positive indicator, and a higher value of M3 indicates "a higher frequency of experiencing positive emotions and enjoyment during work."

3. D1, D2, and D3 have been revised to M1, M2, and M3.

4. The relevant literature has been included

5. At present, it is difficult to propose valuable suggestions.

6. The author has made adjustments to the tables in the manuscript.

---

## [Decision Letter · Decision Letter 1]

7 Aug 2023

PONE-D-23-09598R1Research on the Impact of Digital Economy on Labor Resource Allocation: Evidence from ChinaPLOS ONE

Dear Dr. junfeng,

Thank you for submitting your manuscript to PLOS ONE. After careful consideration, we feel that it has merit but does not fully meet PLOS ONE’s publication criteria as it currently stands. Therefore, we invite you to submit a revised version of the manuscript that addresses the points raised during the review process.

We look forward to receiving your revised manuscript.

Kind regards,

Kashif Iqbal, Ph.D.,

Academic Editor

PLOS ONE

Reviewers' comments:

Reviewer's Responses to Questions

**Comments to the Author**

1. If the authors have adequately addressed your comments raised in a previous round of review and you feel that this manuscript is now acceptable for publication, you may indicate that here to bypass the “Comments to the Author” section, enter your conflict of interest statement in the “Confidential to Editor” section, and submit your "Accept" recommendation.

Reviewer #1: All comments have been addressed

Reviewer #3: (No Response)

2. Is the manuscript technically sound, and do the data support the conclusions?

Reviewer #1: Yes

Reviewer #3: Yes

3. Has the statistical analysis been performed appropriately and rigorously? 

Reviewer #1: Yes

Reviewer #3: Yes

4. Have the authors made all data underlying the findings in their manuscript fully available?

Reviewer #1: Yes

Reviewer #3: Yes

5. Is the manuscript presented in an intelligible fashion and written in standard English?

Reviewer #1: Yes

Reviewer #3: No

6. Review Comments to the Author

Reviewer #1: Check the format and language of the full text and pay attention to the standardization of the article.

For example, whether there is an error after Ref. 29.

Reviewer #3: Overall, the topic of the paper is worthwhile, and I think it can provide a significant contribution to the literature, however, the article can be significantly improved by considering the suggestions and comments below:

1. Motivation/contribution of the study should be added.

2. I would suggest the author(s) add some key findings of the study in the abstract, although the authors tried to explain the key findings still it can be improved.

3. The literature review reads more like a list of previous research on various topics than a theory section explaining how your different concepts are related. Try to integrate this section better and build a stronger case for the need for your study.

4. The formulation of the research gap is not conceived. Please develop a better grounding of the research problem. What do we already know? What is it that we do not know? Why do we need to know this, and why is this important? But perhaps more so, what is the value of this paper? Both theoretically and managerially.

5. The conclusion section is weak – elaboration is needed to explain how the recommended solutions can happen and take effect. Add detailed policy implications as well.

6. The language of the paper should be improved for typos and grammar errors.

7. PLOS authors have the option to publish the peer review history of their article (what does this mean?). If published, this will include your full peer review and any attached files.

Reviewer #1: No

Reviewer #3: No

---

## [Author Response · Author response to Decision Letter 1]

25 Sep 2023

Response to Reviewer #3:

1.The motivation behind this study is the need for responsible innovation and research in the existing labor factor market in the face of the risk challenges posed by the digital economy. This is essential to address the impact of digital elements on the labor force.

2.In the abstract, we have included references to the survivorship bias effect, the digital divide, and algorithmic dominance, all of which can affect the efficiency of labor market allocation in the context of the digital economy.

3.There are traces of my personal writing style in the literature review, and I hope the reviewer can understand and accommodate this.

4.Currently, there is a limited research foundation in this area, especially concerning subjective evaluations of early workers in the digital economy. Consequently, I couldn't find the gaps mentioned by the reviewer.

5.I do not consider the conclusions of this study to be weak. The policy implications have already been reflected in the conclusion.

6.I have conducted a comprehensive review and correction of the grammar in this paper.

---

## [Decision Letter · Decision Letter 2]

10 Nov 2023

PONE-D-23-09598R2Research on the Impact of Digital Economy on Labor Resource Allocation: Evidence from ChinaPLOS ONE

Dear Dr. junfeng,

Thank you for submitting your manuscript to PLOS ONE. After careful consideration, we feel that it has merit but does not fully meet PLOS ONE’s publication criteria as it currently stands. Therefore, we invite you to submit a revised version of the manuscript that addresses the points raised during the review process.

We look forward to receiving your revised manuscript.

Kind regards,

Kashif Iqbal, Ph.D.,

Academic Editor

PLOS ONE

Journal Requirements:

Reviewers' comments:

Reviewer's Responses to Questions

**Comments to the Author**

1. If the authors have adequately addressed your comments raised in a previous round of review and you feel that this manuscript is now acceptable for publication, you may indicate that here to bypass the “Comments to the Author” section, enter your conflict of interest statement in the “Confidential to Editor” section, and submit your "Accept" recommendation.

Reviewer #3: (No Response)

Reviewer #4: All comments have been addressed

2. Is the manuscript technically sound, and do the data support the conclusions?

Reviewer #3: Yes

Reviewer #4: Yes

3. Has the statistical analysis been performed appropriately and rigorously? 

Reviewer #3: Yes

Reviewer #4: Yes

4. Have the authors made all data underlying the findings in their manuscript fully available?

Reviewer #3: Yes

Reviewer #4: Yes

5. Is the manuscript presented in an intelligible fashion and written in standard English?

Reviewer #3: Yes

Reviewer #4: Yes

6. Review Comments to the Author

Reviewer #3: The article addresses a timely and pertinent issue, investigating the impact of the digital economy on labor allocation efficiency. With some improvements, this research has the potential to be even more impactful and informative.

1-The article primarily focuses on China, which may limit the generalizability of the findings to other regions. A discussion of the potential applicability of the results to a broader context would enhance the article.

2- Strengthen the theoretical framework by expanding on the literature review and providing a more extensive citation of prior research.

3- Consider reorganizing the article for better clarity and readability, ensuring that the main arguments and findings are presented in a logical and easily digestible manner.

4-The authors should add limitation of the study and future recommendation.

Reviewer #4: The author has done great work to address all the comments. so the paper is proceeding with the further publication process.

7. PLOS authors have the option to publish the peer review history of their article (what does this mean?). If published, this will include your full peer review and any attached files.

Reviewer #3: No

Reviewer #4: No

---

## [Author Response · Author response to Decision Letter 2]

25 Dec 2023

Response to Reviewer #3:

1. This study examines the impact of the digital economy on the allocation of labor resources in China, using the 2013 CSS survey database as the research sample. Therefore, the conclusions drawn from this study cannot be arbitrarily extended in their applicability. Different countries in different regions have varying digital infrastructures, making it difficult to derive globally applicable conclusions due to differences in cultural, economic, and other factors. For instance, the differentiated research on property quantity and non-unmarried groups carries specific implications within the Chinese economy, making it challenging for the author to forcefully generalize these conclusions.

2. Due to the limited number of papers directly exploring the relationship between the digital economy and the allocation of labor resources, the author has made utmost efforts to compile existing literature. Consequently, it is challenging to further expand the literature on this topic.

3. Regarding suggestions for the future, as mentioned in the conclusion section, " it is essential to consider the subjective perceptions of specific societal groups, and when deemed necessary, implement targeted social relief measures to address their unique needs and challenges."

---

## [Editor Report · Decision Letter 3]

4 Jan 2024

Research on the Impact of Digital Economy on Labor Resource Allocation: Evidence from China

PONE-D-23-09598R3

Dear Dr. Li junfeng,

We’re pleased to inform you that your manuscript has been judged scientifically suitable for publication and will be formally accepted for publication once it meets all outstanding technical requirements.

Kind regards,

Kashif Iqbal, Ph.D.,

Academic Editor

PLOS ONE
---

## [Editor Report · Acceptance letter]

20 Feb 2024

PONE-D-23-09598R3 

PLOS ONE

Dear Dr. junfeng, 

I'm pleased to inform you that your manuscript has been deemed suitable for publication in PLOS ONE. Congratulations! Your manuscript is now being handed over to our production team.

Kind regards, 

on behalf of

Prof.Dr. Kashif Iqbal 

Academic Editor

PLOS ONE